# Detection of Anti-*Neospora caninum* Antibodies on Dairy Cattle Farms in Southern Italy

**DOI:** 10.3390/vetsci9020087

**Published:** 2022-02-17

**Authors:** Raffaella Manca, Giuseppina Ciccarese, Domenico Scaltrito, Daniela Chirizzi

**Affiliations:** Istituto Zooprofilattico Sperimentale della Puglia e della Basilicata (IZS_PB), Via Manfredonia 20, 71121 Foggia, Italy; raffaella.manca@izspb.it (R.M.); giuseppina.ciccarese@izspb.it (G.C.); domenico.scaltrito@izspb.it (D.S.)

**Keywords:** *Neospora caninum*, serodiagnosis, Italian cattle farms

## Abstract

Neosporosis is recognized as one of the major causes of bovine abortion worldwide. Canids are the main definitive host for this parasite and the presence of dogs in the farm is an important factor for the *Neospora caninum* infection in bovines. Since, in the province of Lecce, located in the Apulia region of Southern Italy, there are no studies showing the presence of the infection in farm animals, the objective was to perform a serological evaluation for anti-*N. caninum* antibodiesin serum from 706 dairy cattle and 21 farm dogs located in 40 farms uniformlydistributed over the territory.The presence of *N. caninum* infection was confirmed in 90.0% (36/40) of the 40 farms examined. The results obtained on all serum samples by an enzyme-linked immunosorbent assay (ID Screen^®^
*Neospora caninum* competition ELISA kit) for anti-*N. caninum* antibodies showed a seropositivity rate of 21.1% (149/706) among dairy cows, with a statistically significant higher percentage of positive subjects in the animals over two years old and a positivity rate of 42.9% (9/21) in tested dogs. The obtained data confirmed the presence of neosporosis even in the Lecce area, where it could therefore represent an important cause of abortion and economic losses.

## 1. Introduction

*Neospora caninum* are obligate, intracellular, protozoan parasites of the phylum Apicomplexa that have intermediate hosts (e.g., ruminants) and definitive hosts such as dogs. Infection can cause a clinical disease (neosporosis) that has been recognized as a major cause of reproductive disorders in cows worldwide, including abortion, estrus repetitions, and temporary anestrus [1,2,3,4,5]. Infected cows may abort from 3 months to the end of gestation, although the most *Neospora*-induced abortions occur at 5–6 months [6]. Seroprevalence studies show that infection rates vary considerably between and within countries, and between dairy and beef cattle, and they are estimated to cause substantial economic losses to livestock producers [7,8,9], mainly because effective treatment or vaccination is not yet available. Worldwide surveys have highlighted that *Neospora caninum* is responsible for 15–21% of abortions in cattle [8,10,11]. The parasite is also widespread on cattle farms in Italy with abortion rates comparable to the international literature [12,13,14,15].

*N. caninum* has a heteroxenous life cycle consisting of two distinct modes of reproduction: an asexual reproduction, which occurs in intermediate hosts such as sheep, goats, cattle and others [1,13,16,17,18]; and a sexual reproduction, which only occurs in canids such as dogs (*Canis lupus familiaris*) [19,20,21], coyotes (*Canis latrans*) [22], wolfs (*Canis lupus*) [23,24] and dingoes (*Canis lupus dingo*) [25]. Canids are the definitive host for this parasite, and they may also be affected clinically [26,27,28,29]. They become infected by eating contaminated meat. Oocysts are excreted with the dog’s faeces and persist in the environment for unknown periods of time. Intermediate hosts are infected by ingesting faecally contaminated pasture or water or eating tissues cysts (horizontal or postnatal transmission). Intermediate hosts do not excrete oocysts but can transmit the parasite through the placenta (vertical or transplacental transmission) [30,31,32,33]. A schematic representation of the route of transmission for *N. caninum* is explained in Figure 1.

Although a wide variety of diagnostic tools have been developed [34,35,36,37,38,39,40,41], to date, no definitive serodiagnostic tool for assessing *N. caninum* abortion has been reported [42]. Cases of *Neospora* abortion can be confirmed through detection of *N. caninum* tachyzoites in fetal or maternal lesions, while demonstration of specific antibodies in maternal sera or fetal fluids provides strong evidence that abortion might be associated with Neosporosis. Detection of *N. caninum* antibodies can be achieved by many serological tests, including immunofluorescence antibody test (IFAT), immunostimulating complex enzyme-linked immunoassay and enzyme-linked immunosorbent assay (ELISA). Although IFAT is one of the most reliable serological tests for detection of *Neospora* antibodies, high cost, extensive consumption time and the need for specialized equipment and expertise have limited its use [43]. Currently, enzyme-linked immunosorbent assays (ELISAs) are the most widely marketed serological tests. They are suitable for the detection of *Neospora* infection in large-scale surveillance studies because they are low cost and fast to perform. [14,17,44,45,46,47].

In Italy, the prevalence of neosporosis and the relationship between *N. caninum* infections in dogs and cattle was analyzed with both serological and molecular methods [12,13,25,48,49,50,51,52]. In Italy, as no official control plans for neosporosis have never been implemented, prevalence data can be mainly deduced from local studies [53]. In Northern Italy, the monitoring of infectious abortions revealed data in agreement with the international literature [54,55,56,57,58], showing percentages of seropositivity ranging from 11.5% to 29.3% in aborted foetuses and percentages of seropositivity between 35.5% and 40.5% in cows that had aborted [59]. In Veneto an average seroprevalence of 22.0% was reported in 1925 dairy cows and 43.2% of these animals had aborted [60]. In Sardinia, surveys conducted through IFAT found prevalence rates of 64.0% in dairy cattle [61], while positive farms were found to be 55.0% by immunostimulating complex enzyme-linked assay on bulk milk [52]. While in Sicily the seroprevalence in dairy cattle was 26.2% by ELISA, and the prevalence among different farms examined was 77.8% [62]. Few studies have been conducted regarding the diffusion of *Neospora* in Southern Italy [12,46,63] and no data were found in literature regarding the diffusion of *N. caninum* in the province of Lecce in dairy cows and in farm dogs.

In this study a serological evaluation against *N. caninum* was carried out on dairy cattle farms uniformly located over the province of Lecce. Dairy cattle under two years old and offspring of positive subjects were tested for *N. caninum* to better understand the role and efficiency of vertical transmission. Instead, dairy cattle over two years old were tested to better understand the role of horizontal transmission in the spread of neosporosis in herds. In addition, dogs were analyzed for neosporosis on the farms where they were present to assess whether there was a correlation between the immune status of dogs and cattle against this disease.

## 2. Materials and Methods

### 2.1. Institutional Animal Care and Use Committee

Animal blood sampling was based on the 2010/63 EU guidelines of European Community and Council on the Protection of Animals used for Scientific Purposes; according to Directive Article 1, Paragraph 5, Element F, “practices not likely to cause pain, suffering, distress or lasting harm equivalent to, or higher than, that caused by the introduction of a needle in accordance with good veterinary practice are permitted for experimental purposes.”

### 2.2. Sampling Procedures

Data collected between 2017 and 2021 in 40 dairy farms evenly distributed in the province of Lecce were analysedin order to have a geographical evaluation of the presence of the *N. caninum* infection on the territory (Figure 2). Among the farms, 38 presented no relevant evidence of reproduction problems (no-RDF), while two farms showed a high incidence of reproductive disorders (RDF) that could be due to neosporosis.

We collected blood samples of 766 cows and 21 farm dogs. Cows were divided according to age into two groups—animals under and animals over two years of age—to better understand the distribution of seropositive samples in function of age. For each farm, at least 10 animals over 2 years old and at least 5 animals under 2 years old were randomly selected. Among all cows, 706 (262 animals under two years old and 444 ones over two years old) came from 38 no-RDFs and 60 (31 animals under two years old and 39 ones over two years old) from two RDFs. In addition, ten cows were selected among those resulted seropositive during the first year of examination and these subjects were retested annually for three consecutive years to assess the persistence of seropositivity over time. Moreover, ten offspring of seropositive subjects were tested to assess transplacental transmissibility. These subjects were aged between 6–12 months to limit interference from maternal antibodies obtained through colostrum intake [60].

Concerning the sampling of dogs, no distinction was made regard breed, sex and age. At least one canine blood sample from 18 farms was collected. The only requirement considered was that the dog belonged to the farms under study. Of the 21 dogs, 19 came from 17 no-RDFs and 2 from one of RDFs. In other 14 farms no dogs could be sampled because they were aggressive or uncontrollable and, finally, in 8 farms no dogs were present. Regarding the two RDFs, only one had two dogs and of these only one had anti-*N. caninum* antibodies.

Blood samples were collected from the caudal vein in cows and cephalic vein in dogs and stored in clot activator tube and transported under refrigeration (+4 °C) to the laboratory. After collection, blood was centrifuged to 3000 rpm (1800× *g*) for 10 min and separated sera were stored at −20 °C until examination.

### 2.3. Serological Testing and Procedures

All serum samples were tested for antibodies against *N. caninum* by a competitive enzyme-linked immunosorbent assay (ELISA) using commercial test kit based on purified extract of *N. caninum* (ID Screen^®^
*Neospora caninum* competition by IDVet Innovative Diagnostics, Grabels, France) and following the manufacturer’s instructions. The selected kit was validated for the detection anti-*N. caninum* antibodies in serum or plasma from ruminants, dogs, or other susceptible species. All sera were distributed in duplicate and 50 µL from each sample were distributed in slides in different wells and were incubated for 45 min at 37 °C in a humid chamber. After three rinses with buffer wash, 100 µl conjugate was added to each well. Samples were incubated at 5 °C for 30 min followed by three rinses with buffer wash. Afterwards 100 µL of substrate solution was added to each well and left at 5 °C for 15 min and, finally, 100 µL of stop solution was added. The corresponding values for optical density were recorded by a 96-well microplates reader (Spectro Star Omega-BMG Labtech) at a wavelength of 450 nm by software Omega 5.70. The results were calculated as the percentage S/N (S/N%). A test result of ≤50 S/N% indicated a positive result, 50% < S/N% ≤ 60 indicated a doubtful result and >60 S/N% a negative result. Specific details of the ELISA kits, along with the sensitivities and specificities of the assays, are 100% (IC 95%: 98.8–100%) and 100% (IC 95%: 99.41–100%), respectively.

### 2.4. Statistical Analysis

The difference of seropositivity for *N. caninum* of the epidemiological data according to the statistically significant *p* value of 0.05 or 0.01 were evaluated by the Chi-square (*χ*^2^) test by software R-4.1.2 for Windows. Four evaluations were conducted in parallel on cattle: (a) if in no-RDFs the seropositivity of *N. caninum* of cows under two years old was statistically higher than in cows over two years old; (b) if the seropositivity of *N.caninum* in RDFs was statistically higher than in no-RDF ones; (c) if the seropositivity rate was statistically higher in cattle less than two years old on RDFs than on no-RDFs. (d) if in RDFs the seropositivity of *N. caninum* of cows under two years old was statistically higher than in cows over two years old. In addition, correlations between seropositivity in dogs and cattle were analyzed.

#### 2.4.1. Data Set Point (a)

All samples from the two farms considered to be RDF and samples found to be doubtful in the ELISA test were excluded from statistical analysis. Remaining data (700) were divided according to age into two groups discarding: animals under two years old (group A) and animals over two years of age (group B). The data obtained in groups A (260 samples) and B (440 samples) were subjected to statistical analysis.

#### 2.4.2. Data Set Point (b)

Out of all the sample data from the 40 farms, samples (7) found to be doubtful in the ELISA test were excluded from the statistical analysis. The remainder (759) were divided according to the type of farm (RDFs and no-RDFs) into seropositive (group C) and seronegative (group D), respectively. The data obtained in groups C (187 samples) and D (572 samples) were analyzed statistically.

#### 2.4.3. Data Set Point (c)

All data on cattle under two years old (298) were subdivided according to the type of farm of origin (RDFs and no-RDFs) into seropositive (group E) and seronegative (group F), respectively and were analyzed statistically. No cattle under two years of age were found to be doubtful in the ELISA test. The data obtained in groups E (38 samples) and F (260 samples) were analyzed statistically.

#### 2.4.4. Data Set Point (d)

The sample found to be doubtful in the ELISA test was excluded from statistical analysis. Remaining data (59) were divided according to age into two groups discarding: animals under two years old (group G) and animals over two years of age (group H). The data obtained in groups G (38 samples) and H (21 samples) were subjected to statistical analysis.

## 3. Results

### 3.1. Detection of Anti-Neospora caninum antibodies in Dairy Cattle 

The presence of *N. caninum* infection in cows was found in 36 out of 40 farms (90.0%) examined. The data resulting from the tests carried out in no-RDFs and RDFs were reported in Table 1. 

The seropositivity rate for *N. caninum* in cattle belonging to no-RDFs (38/40) was 21.1% (149/706). A total of 149 seropositive, 6 doubtful and 551 seronegative animals were identified. On the other hand, a high seropositivity rate of 63.3% (38/60) was found in the two RDFs with 34.2% (13/38) of cattle older than two years and 65.8% (25/38) younger than two years found seropositive.

The tests carried out on ten offspring, male or female, born from seropositive subjects showed that 80% had anti-*N. caninum* antibodies. In addition, persistence of seropositivity for *N. caninum* was observed for all three years in the ten positive cows retested annually.

### 3.2. Statistical Analysis

The data sets were subjected to statistical analysis using the x^2^-test, as resumed in Table 2, Table 3, Table 4 and Table 5. The results of groups A (no-RDFs animals under two years old) and B (no-RDFs animals over two years old) showed a significant difference between the two groups with the highest number (105/700) of positive subjects in the group of cattle older than two years of age (*p* < 0.05). While the data set of groups C (seropositive animals) and D (seronegative animals) showed that the difference in seropositivity for *N. caninum* in RDFs (149/759) is statistically higher than in no-RDF ones (*p* < 0.01). The data sets of groups E (seropositive cattle under two years old) and F (seronegative cattle under two years old) showed that the seroprevalence (44/298) was statistically higher in cattle less than two years old on RDFs than on no-RDF ones (*p* < 0.01). Finally, the results of groups G (RDFs animals under two years old) and H (RDFs animals over two years old) showed an insignificant difference between the two groups. Furthermore, no significant correlations were found between the presence of seropositive dogs and seropositivity of herds of the same farm.

### 3.3. Detection of Anti-Neospora caninum Antibodies in Farm Dogs 

The results of the ELISA tests, shown in Figure 3, revealed the presence of *N. caninum* antibodies in 9 out of 21 farm dogs tested, which corresponds to a seropositivity rate of 42.9% (9/21). In addition, in half of the farms tested (9/18) at least one dog was found to be seropositive for *N. caninum*.

## 4. Discussion

The ELISA method allowed a rapid and inexpensive screening of neosporosis presence in dairy farms. The result of this study highlights that *N. caninum* infection is very common in dairy farms in Lecce because 90% of farms have seropositive animals. Moreover, it cannot be excluded that farm in which *N. caninum* antibodies has not been detected, are completely free of infection because it has not been possible to test all animals. 

The seropositivity rate of 21.1% found in dairy cattle was similar to or higher than that reported in other surveys in Italy [14,37,46]. Among the dairy cows tested, animals older than two years showed statistically higher seropositivity than those less than two years of age. These results agree with previous studies [64,65], in which it was shown that an animal’s chance of being positive for *N. caninum* increases with age due to increased risk of infection related to horizontal transmission [18,41,66]. The persistence of positivity in animals over time shows that animals once in contact with the parasite, remain infected for life with the possibility of clinical manifestations such as abortion based on the balance that is created between the immune system of the host and the replicative capacity of the parasite [4,5,24]. The role of vertical transmission was reaffirmed since 80.0% of female and male offspring of seropositive subjects were reactive to the *N. caninum* test. A significant association was shown between serostatus of offspring/mothers in agreement with the literature [65,67]. These observations indicated that congenital infection was an important transmission route. The finding that offspring may be seropositive or seronegative, suggests that maternal rather than fetal factors probably determined the transmission of the infection to the fetus [65,68]. Seropositive progeny had more than a three-fold increased risk of abortion compared with seronegative offspring [69]. This suggests an enormous impact of congenital *N. caninum* infection on abortion risk. Therefore, it seems prudent to exclude congenitally infected calves as replacement stock, as suggested previously [68], in order to decrease the future risk of abortion in the herd and to discontinue vertical transmission over the generations. This aspect has been highlighted by many authors who point out that the vertical route is the main transmission route of *N. caninum* in cattle, estimated at an average of 75 to 90.0%, without forgetting the risk of horizontal transmission due to ingestion by the cattle of oocysts eliminated with faeces by the definitive hosts (dogs, dingoes, coyotes, wolfs) [18,19,21].

Regarding the results obtained in RDFs, a positivity rate of 63.3% was found both adults and youth. This result, evidently higher than that found in the remaining 38 farms examined, may be related to the different routes of transmission of neosporosis. In RDFs, the disease manifested itself in an epidemic way, while in the others, which did not show evident reproductive disorders, it was probably present in a sporadic or endemic form [7]. Although the number of subjects tested was not very high, dogs also presented high seropositivity rates for *N. caninum*. The involvement of dogs in the life cycle of the parasite, and the possibility that this species acts as an amplifier of the disease is well known [16,17,18,28]. Different studies have found positive associations between the seropositivity to *N. caninum* in cattle and the presence [70,71], or number [12,72], or density [73] of dogs on farms. Contrary to previous reports [18,46,71,74], in this study no relationship was found between the presence of seropositive dogs and the seropositivity of herds. This is probably due to the small number of cattle and dogs sampled, but the detection of positive cattle even on farms where no dogs were present demonstrates that the infection of cattle starts with the dog, but proceeds and spreads independently of the dog via the vertical transmission route. 

## 5. Conclusions

This was the first epidemiological study to evaluate the presence of *N. caninum* infection in dairy farms in the province of Lecce, located in the Apulia region of Southern Italy, evidencing that neosporosis infection is widespread throughout this area in both cattle and dogs. In fact, this study showed the presence of a seropositive animal in 90.0% of the farms and in 42.9% of farm dogs tested. Statistical analysis evidenced that the possibility of a cattle to be positive increases with age due to an increased risk of horizontal transmission. Identification of high level of seropositivity in RDF farms shows that *N caninum* may be involved, confirming that neosporosis is an important cause of reproductive disorders also in this territory. Considering the absence of local official disease control plans, information programs for farmers on neosporosis and the effects it can have should be applied, in order to implement good herd management practices such as limiting the presence of stray dogs and removing positive animals from the herd with the aim of reducing the incidence and spread of this disease.

## Figures and Tables

**Figure 1 vetsci-09-00087-f001:**
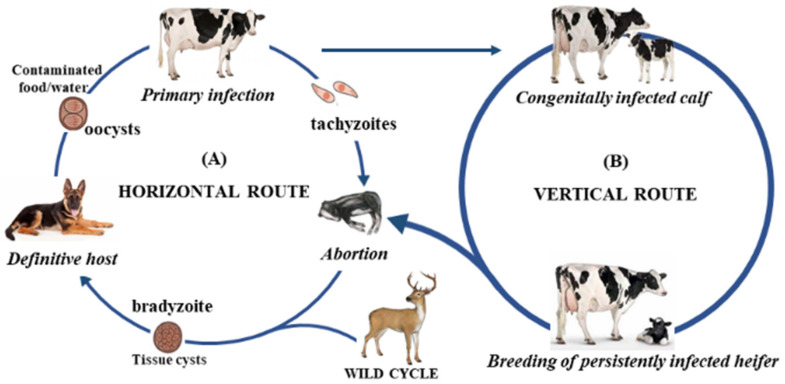
Horizontal (**A**) and vertical (**B**) transmission routes of *N. caninum*.

**Figure 2 vetsci-09-00087-f002:**
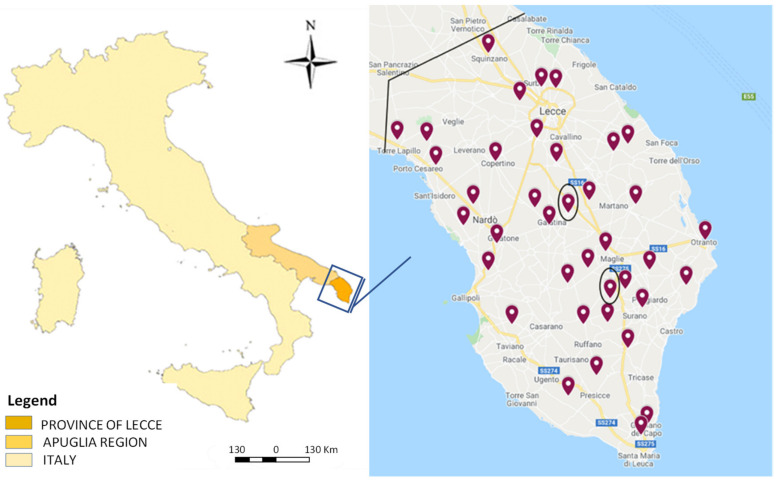
Geographical location of the 40 dairy farms in Lecce territory involved in this study from 2017 to 2021.The two farms with reproductive disorders are indicated with black bordered markers.

**Figure 3 vetsci-09-00087-f003:**
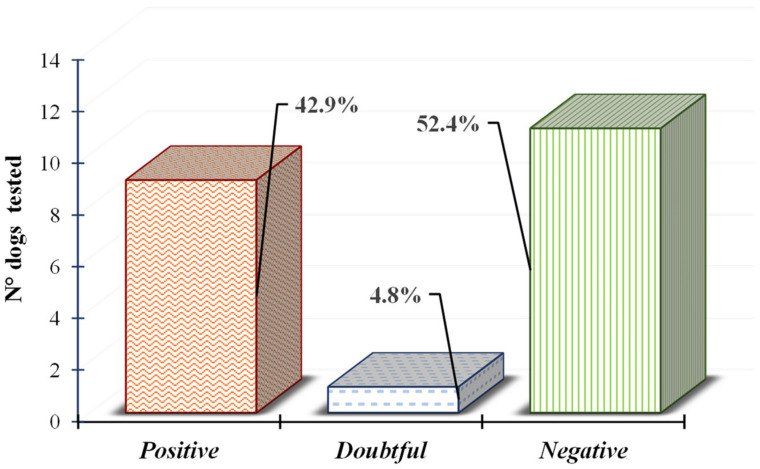
Distribution of serological results (positive in red, doubtful in grey and negative in green) for antibodies against *N. caninum* in farm dog serum.

**Table 1 vetsci-09-00087-t001:** Distribution of seropositivity for *N. caninum* among tested cows in dairy cattle farms in Lecce province between 2017 and 2021.

FARMS	TOT	<2 years	>2 years	Positive	Doubtful	Negative
no-RDF *	706	262 (37.1%)	444 (62.9%)	149 (21.1%)	6 (0.8%)	551 (78.1%)
RDF **	60	39 (65.0%)	21 (35.0%)	38 (63.3%)	1 (1.7%)	21 (35.0%)

* no-RDF = no relevant evidence of reproduction problems; ** RDF = high incidence of reproductive disorders.

**Table 2 vetsci-09-00087-t002:** Numbers of seropositive samples for *N. caninum* in no-RDFs between groups A (240 samples) and B (440 samples). a, b: *p* < 0.05.

Dairy Cattle Samples	Positive to ELISA	Negative to ELISA
<2 years (A)	44 ^(a)^	216
>2 years (B)	105 ^(b)^	335

x^2^ = 4.669.

**Table 3 vetsci-09-00087-t003:** Numbers of seropositive samples for *N. caninum* between groups C (60 samples) and D (700 samples). c, d: *p* < 0.01.

Farms	Positive to ELISA Test	Negative to ELISA Test
RDF (C)	38 ^(c)^	21
no-RDF (D)	149 ^(d)^	551

x^2^ = 54.492.

**Table 4 vetsci-09-00087-t004:** Numbers of dairy cattle samples <2 years old and positive for *N. caninum* between groups E (38 samples) and F (260 samples). e, f: *p* < 0.01.

Farms	Positive to ELISA Test	Negative to ELISA Test
RDF (E)	25 ^(e)^	13
no-RDF (F)	44 ^(f)^	216

x^2^ = 44.495.

**Table 5 vetsci-09-00087-t005:** Number of seropositive samples for *N. caninum* in RDFs between groups G (38 samples) and H (21 samples). g, h: *p* < 0.05.

Dairy cattle samples	Positive to ELISA Test	Negative to ELISA Test
<2 years (G)	25 ^(g)^	13
>2 years (H)	13 ^(h)^	8

x^2^ = 0.089.

## Data Availability

The data presented in this study are all available within the article.

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
