# Peer review of "Detection of Anti-Neospora caninum Antibodies on Dairy Cattle Farms in Southern Italy"

_vetsci, 2022, doi:10.3390/vetsci9020087_

Round 1
Reviewer 1 Report
The manuscript vetsci-1540172 " Detection of antibodies against N. caninum in serum from cows and dogs in dairy cattle farms in South Italy " by Raffaella Manca et al provides the data of serological epidemiology on infections by N. caninum in cows and dogs in dairy cattle farms in South Italy. Although of interest, I feel that the manuscript requires major revision before it is suitable for publication.
Major comments:
1.The significance of seroepidemiological detection of N. caninum in dogs should be further elaborated.
2.How many are there the detection methods of infection by N. caninum in dogs? Why did not they compare the serological detection with detection of oocyst or PCR for dog feces?
3.The procedure of the manufacturer’s instruction (ID ScreenR Neospora caninum by IDVet Innovative Diagnostics, Grabels, France) shall be briefly introduced.
4.The section of preface and conclusion should be simplified.
Author Response
Replies to Referee 1
1) “The significance of seroepidemiological detection of N. caninum in dogs should be further elaborated.”
We thank the Reviewer for his suggestion. The significance of the seroepidemiological detection of N. caninum in dogs has been further elaborated”.
2) “How many are there the detection methods of infection by N. caninum in dogs? Why did not they compare the serological detection with detection of oocyst or PCR for dog feces?”
In our work, N. caninum infection in dogs was only assessed by ELISA tests. This, compared to molecular biology investigations, is the fastest and cheapest method that can be applied in breeding.
3) “The procedure of the manufacturer’s instruction (ID ScreenR Neospora caninum by IDVet Innovative Diagnostics, Grabels, France) shall be briefly introduced.”
We thank the Reviewer for his suggestion. The authors have added a description of the Elisa procedure in the corrected version of the manuscript (see lines 130-141).
4) “The section of preface and conclusion should be simplified”
We thank the Reviewer for his suggestion. The preface and conclusion sections have been revised and simplified.
Welcoming Reviewer’s suggestion, we have checked the style of the manuscript and we hope that the quality of English language in the revised manuscript can be considered good.
Reviewer 2 Report
The authors studied the seroprevalence of Neospora caninum in dairy cattle farms in South Italy. The manuscript must be widely improved. First of all, the English should be corrected by a native speaker. The references (particularly those about previous seroprevalence studies in Italy) should be improved. It is only a seroprevalence study and the authors try to search conclusions with are not supported by their results. Going line by line my comments are the following:
Line 1: title: change detection of abs against … by “Detection of anti-Neospora caninum antibodies in …”
Line 5: letter size
Line 10: (N. caninum) should be removed.
Line 10-11: Remove this sentence “N. caninum has been detected in every country in which 10 a study has been specifically conducted to detect this parasite in cattle” since you said worldwide in line 7.
Line 12: 706 dairy cows instead of dairy cattle. How many farms?
Line 13: “considering that in this territory there are no studies showing the presence of the infection in these animals” please move this sentence before to the explanation of the animals tested. Considering that…, the objective was…
Line 13: What is Lecce?
Line 14: (X/40)
Line 15: You should give a brief detail about the ELISA used.
Line 15: change positivity by seropositivity in all the manuscript.
Line 17: add a solid and contrasted conclusion.
Line 18: Keywords, add “serodiagnosis” and “Italy”
Line 22: “wide host range”, please be more specific (cattle, sheep, goats, ect) and explain the definitive/intermediate hosts.
Line 24: be more specific about the main term of gestation in which the abortions occur. Please, try to cite Dubey et al., 2017 (book, Neosporosis in animals).
Lines 24-28: “Other infections that 24 can cause abortion also exist, such as brucellosis [4-5], leptospirosis [6] and pestivirus [7], 25 but N. caninum is increasingly identified as the most common abortifacient [8-9].”. This sentence should be removed from the manuscript since it does not provide essential information for your study.
Line 30: Try to look for in the book “Neosporosis in animals”, Dubey et al., 2017. Cattle Seroprevalence in tables.
Line 31: You did not included all the seroprevalence studies in Italy, please see Pubmed:
https://pubmed.ncbi.nlm.nih.gov/16323023/
and many other seroprevalence studies in Italy
Please you should also say something about the molecular diagnosis in your country
https://parasitesandvectors.biomedcentral.com/articles/10.1186/s13071-020-04557-6
https://pubmed.ncbi.nlm.nih.gov/32157394/
Lines 32-34: these sentences should be removed since in your study you did not apply any control meaure. At least you must avoid repetitions (already said the control options in lines 56-58).
Line 36: modes of reproduction? In the figure you stated the modes of transmission not the modes of reproduction. Please be careful with the nomenclature. Figure 1, minor and principal routes should be changed by horizontal and vertical.
Line 40: add small ruminants (sheep, goats) as IH and delete pigs and birds since these hosts are absolutely anecdotic.
Lines 41-42: “In cattle, N. caninum is generally viewed as primarily an abortifacient and abortions follow three main patterns: sporadic, endemic and epidemic abortions [11].” Please rewrite this sentence.
Lines 43-45: add any reference in this sentence.
Line 45: “for the first time early in the pregnancy” ¿? Please confirm it and rewrite.
“for the most part” ¿? correction by a native speaker is essential.
Lines 46-48: “The only reported physical sign in serologically positive adults has been weight loss in beef cattle [32].” Remove it since it has not relevance in the pathology of this disease and even less relevance for your study.
Line 49: repercussions? Correction by a native English speaker. Outcome?
Lines 49-54: in this study you did not evaluate the fetal N. caninum infection so this paragraph must be removed from the introduction.
Lines 54-56: apart from the lesions you should also cite other methologies for diagnosis of Nc infection such as molecular diagnosis etc.. with its pros and cons of all the methodologies, including serology.
Lines 73-76: “Considering the impact that N. caninum has on dairy cattle breeding and the absence of valid vaccines and principals for the treatment of the infection, the importance of knowledge on the epidemiology of this parasite to implement effective measures for its control is evident”. You should remove it since you did not apply any control measure in your study.
Line 81: rewrite the objective to improve the understanding, mainly concerning dog serology.
Line 83: transmissibility? Better “efficiency of vertical transmission”.
Lines 82-84: Why not to test the progeny of seronegative cows to evaluate the presence of horizontal transmission? In your study maybe it has more sense than the testing of the dogs.
Lines 89-90: “The territory is mainly flat with 281 km of coastline mostly low and sandy with jagged edges, which are followed by rough and rocky stretches and cliffs overlooking the sea”. Please delete, since it is not important for your study.
Lines 100-111. The number of animals, farms and the selection of the animals should be placed BEFORE the methodology used the serological evaluation.
Lines 102-104: Which is the cut off in the abortion rate to consider a farm with/without reproductive disorders? Do you have reliable reproductive data? High incidence?
Line 104: if you consider that this 2 farms suffered abortion outbreaks, why do you not complemented the study with PCR of the fetuses and avidity tests?
Lines 91-92: vacuum tubes? Supplier? Volume? Centrifugation to obtain serum (speed and time)? Storage in the laboratory?
Line 97: Brand of the ELISA reader.
Lines 98-99: Se and Sp are near 100% but not exactly 100%. Please see Table 2 of this article (Serological diagnosis of bovine neosporosis: A comparative study of commercially available ELISA tests. Alvarez-García et al., 2013). Consider the citation of this article in your manuscript. Are the IdVET ELISA kit you used to test dogs validated for dog sera? Did you used more than one IDVet ELISA kit?
Line 96: “following the manufacturer’s instructions”. Please, you should write a brief description of the protocol (OD, nm, formula for the S/P% and the cut-off).
Lines 120-124: the whole paragraph contains material and methods and have to be removed from the results section.
Line 126: title, “anti- N. caninum antibodies”
Line 127: in how many farms did you detect antibodies? (X/40)
Lines 128-130: this sentence have to be moved to the discussion section.
Lines 130-132: this sentence should be rewrite, placing the seropositive rate at the beginning ot the sentence (149/706). Since the Se and Sp are not 100%, you should calculate the true seroprevalence rate.
Table 1: should be deleted. Figure 3 is enough to show the seroprevalence in no-SOF farms. The SOF farms are explained below (consider the option of put SOF and no SOF results in the same section). You should answer this question, is the seroprevalence in SOF farms statistically higher than in no-SOF ones?
“Dubious” better “doubtful”
Line 153: figure 4 must be removed since data of this pie chart are material and methods (no results).
Line 150: “the text continues here” should be removed in every part of the manuscript in which the authors wrote it.
Lines 157-160: the statistical analysis have to be explained in the material and methods section (and delete this “p-value=0.03018017 so P<0,05” from the results). You should answer this question, are there statistically higher seroprevalence rate in less than two years old cattle from SOF farms compared to non-SOF farms?
Lines 161-163: you should also calculate the odds ratio.
Lines 172-173: “In this study, were tested 10 calf under one year old, which were male or female 172 offspring of N. caninum. positive cows”. This sentence is material and methods (no results).
Line 173: maternal antibodies obtained thorough colostrum intake can be detected for several months in the offspring. Please clarify the age of the ofsspring tested and evaluate the possibility of the maternal interference.
Line 175: you should detail the variation in the S/P% of these animals along years. It is not enough the maintenance of the seropositivity. Increased S/P% could mean recrudescence of the chronic infection.
Line 178: high incidence? Which incidence? Objective data.
Lines 178-187: all these sentences are not results and they must be moved (avoiding repetition) to the material and methods section.
Lines 188-190: X% (13/38) X% (25/38). Youth? Please the English should be corrected by a native speaker.
“This result, evidently higher than that found in the remaining 38 farms examined, is related to the different modes of manifestation of neosporosis” it is not true at all. You must complement your study with additional evaluations to know the predominant route of infection. With your results you only can hypothesize it in the discussion section? And please see if you can do this (are there statistically higher seroprevalence rate in less than two years old cattle from SOF farms compared to non-SOF farms?).
Lines 192-193: “The presence of dogs was not evidenced on all farms. Among examined farms, were 192 collected blood samples of 21 dogs”. This should be moved to the material and methods section instead of results section.
Lines 193-196: if the methology used to test antibodies in dogs has not been validated these results are invalid. In my opinion you should remove the serology of the dogs from the manuscript. The dog seropositivity is not indicative of horizontal transmission since many sero+ dogs have never shed oocysts.
Lines 207-217: rewrite to improve the understanding. Many ideas without a guiding line (avoid repetition of information). You must compare you results on the seroprevalence of Neospora in South Italy with previous studies in this country, taking into account the different methologies used.
Lines 210-211: correct citation of the references.
Lines 205-217: your hypothesis of horizontal tranmssion should be confirm with solid data/test/analysis.
Lines 218-219: influence of maternal antibodies in the offspring results?
Lines 241-262: please see my recommendations about dog serology.
Lines 273-278: you did not apply any control measure so this must be removed from the manucript.
Lines 279-289: many mixed and inconsistent ideas. Please rewrite.
Line 293: farm seropositivity not seropositivity level.
Lines 296-299: you did not show the number of aborted/repeteadly aborted animals, so this should be removed.
Line 301: you did not study the abortions and the economic impact and therefore it should be removed.
Lines 303-312: remove since you did not apply any control measure. Please give only conclusions about the serology not about abortions, economic impact and control programs.
Author Response
Replies to Referee 2
Line 1: title: change detection of abs against … by “Detection of anti-Neospora caninum antibodies in …”
We thank the Reviewer for his suggestion and, if the article will be accepted for publication, the title will be changed: “Detection of anti-Neospora caninum antibodies in serum from cows and dogs in dairy cattle farms in South Italy”.
Line 5: letter size
We thank the Reviewer for his observation. The authors adjusted the font size.
Line 10: (N. caninum) should be removed.
We thank the Reviewer for his observation. The authors have moved (N. caninum) to line 23 of the revised manuscript.
Line 10-11: Remove this sentence “N. caninum has been detected in every country in which 10 a study has been specifically conducted to detect this parasite in cattle” since you said worldwide in line 7.
We thank the Reviewer for his suggestion. The authors have removed the sentence.
Line 12: 706 dairy cows instead of dairy cattle. How many farms?
We thank the Reviewer for his observation. The authors have clarified the number of farms (40).
Line 13: “considering that in this territory there are no studies showing the presence of the infection in these animals” please move this sentence before to the explanation of the animals tested. Considering that…, the objective was…
We thank the Reviewer for his observation. The authors have moved the sentence.
Line 13: What is Lecce?
Lecce is a province located in the Apulia region of southern Italy.
Line 14: (X/40) and Line 127: in how many farms did you detect antibodies? (X/40)
We thank the Reviewer for his observation. The authors added the number of seropositive farms (36/40) as requested.
Line 15: You should give a brief detail about the ELISA used.
We thank the Reviewer for his suggestion. The authors have added a description of the Elisa procedure in the corrected version of the manuscript
Line 15: change positivity by seropositivity in all the manuscript.
We thank the Reviewer for his suggestion. The authors have changed the description in the corrected version of the manuscript.
Line 17: add a solid and contrasted conclusion.
We thank the Reviewer for his suggestion. The authors have provided to add a solid and contrasted conclusion.
Line 18: Keywords, add “serodiagnosis” and “Italy”
We thank the Reviewer for his suggestion. The authors have cabled the keywords (Neospora caninum; Serodiagnosis; Italian farms). Maximum number allowed was three.
Line 22: “wide host range”, please be more specific (cattle, sheep, goats, ect) and explain the definitive/intermediate hosts.
We thank the Reviewer for his suggestion. The authors have explicated “wide host range” as required.
Line 24: be more specific about the main term of gestation in which the abortions occur. Please, try to cite Dubey et al., 2017 (book, Neosporosis in animals).
Line 30: Try to look for in the book “Neosporosis in animals”, Dubey et al., 2017. Cattle Seroprevalence in tables.
We thank the Reviewer for his suggestion. The authors have added the reference (Dubey et al., 2017. Book, Neosporosis in animals) in the corrected version of the manuscript.
Line 31: You did not included all the seroprevalence studies in Italy, please see Pubmed:
https://pubmed.ncbi.nlm.nih.gov/16323023/
and many other seroprevalence studies in Italy
Please you should also say something about the molecular diagnosis in your country
https://parasitesandvectors.biomedcentral.com/articles/10.1186/s13071-020-04557-6
https://pubmed.ncbi.nlm.nih.gov/32157394/
We thank the Reviewer for his suggestion. The authors have revised and updated the bibliography.
Lines 32-34: these sentences should be removed since in your study you did not apply any control meaure. At least you must avoid repetitions (already said the control options in lines 56-58).
We thank the Reviewer for his observation. The authors have moved the sentence.
Line 36: modes of reproduction? In the figure you stated the modes of transmission not the modes of reproduction. Please be careful with the nomenclature. Figure 1, minor and principal routes should be changed by horizontal and vertical.
We thank the Reviewer for his observation. The authors have changed the figure 1.
Line 40: add small ruminants (sheep, goats) as IH and delete pigs and birds since these hosts are absolutely anecdotic.
We thank the Reviewer for his suggestion. The authors have updated the sentence.
Lines 41-42: “In cattle, N. caninum is generally viewed as primarily an abortifacient and abortions follow three main patterns: sporadic, endemic and epidemic abortions [11].” Please rewrite this sentence.
We thank the Reviewer for his suggestion. The authors rewrote the sentence.
Lines 43-45: add any reference in this sentence.
We thank the Reviewer for his suggestion. The authors have added more reference in the sentence.
Line 45: “for the first time early in the pregnancy” ¿? Please confirm it and rewrite.
We thank the Reviewer for his suggestion. The authors rewrote the sentence.
Lines 46-48: “The only reported physical sign in serologically positive adults has been weight loss in beef cattle [32].” Remove it since it has not relevance in the pathology of this disease and even less relevance for your study.
We thank the Reviewer for his suggestion. The authors have removed the sentence.
Line 49: repercussions? Correction by a native English speaker. Outcome?
We thank the Reviewer for his observation. The authors have replaced repercussions with consequences.
Lines 49-54: in this study you did not evaluate the fetal N. caninum infection so this paragraph must be removed from the introduction.
We thank the Reviewer for his suggestion. The authors have removed the sentence.
Lines 54-56: apart from the lesions you should also cite other methologies for diagnosis of Nc infection such as molecular diagnosis etc.. with its pros and cons of all the methodologies, including serology.
We thank the Reviewer for his suggestion. The authors have mentioned the different survey tools available and updated the bibliography.
Lines 73-76: “Considering the impact that N. caninum has on dairy cattle breeding and the absence of valid vaccines and principals for the treatment of the infection, the importance of knowledge on the epidemiology of this parasite to implement effective measures for its control is evident”. You should remove it since you did not apply any control measure in your study.
We thank the Reviewer for his suggestion. The authors rewrote the sentence.
Line 81: rewrite the objective to improve the understanding, mainly concerning dog serology.
We thank the Reviewer for his suggestion. The authors rewrote the sentence.
Line 83: transmissibility? Better “efficiency of vertical transmission”.
We thank the Reviewer for his suggestion. The authors have replaced “transmissibility” with “efficiency of vertical transmission”.
Lines 82-84: Why not to test the progeny of seronegative cows to evaluate the presence of horizontal transmission? In your study maybe it has more sense than the testing of the dogs.
We thank the Reviewer for his observation. Offspring from seronegative cows were not tested because the authors did not have sufficient blood samples from those cows.
Lines 89-90: “The territory is mainly flat with 281 km of coastline mostly low and sandy with jagged edges, which are followed by rough and rocky stretches and cliffs overlooking the sea”. Please delete, since it is not important for your study.
Lines 100-111. The number of animals, farms and the selection of the animals should be placed BEFORE the methodology used the serological evaluation.
We thank the Reviewer for his observation. The authors have moved the sentence.
Lines 102-104: Which is the cut off in the abortion rate to consider a farm with/without reproductive disorders? Do you have reliable reproductive data? High incidence?
We thank the Reviewer for his observation. The authors, in order to classify a farm as a suspected outbreak, anamnestic and clinical aspects with a positivity cut-off of more than 40% were considered.
Line 104: if you consider that this 2 farms suffered abortion outbreaks, why do you not complemented the study with PCR of the fetuses and avidity tests?
We thank the Reviewer for his observation. Work in progress on this aspect (data were still limited).
Lines 91-92: vacuum tubes? Supplier? Volume? Centrifugation to obtain serum (speed and time)? Storage in the laboratory?
We thank the Reviewer for his suggestion. The authors have included a description of the sampling procedure in the corrected version of the manuscript.
Line 97: Brand of the ELISA reader.
We thank the Reviewer for his observation. The authors have included the brand of the ELISA reader and software used (Spectro Star Omega-BMG Labtech. Software Omega 5.70).
Lines 98-99: Se and Sp are near 100% but not exactly 100%. Please see Table 2 of this article (Serological diagnosis of bovine neosporosis: A comparative study of commercially available ELISA tests. Alvarez-García et al., 2013). Consider the citation of this article in your manuscript. Are the IdVET ELISA kit you used to test dogs validated for dog sera? Did you used more than one IDVet ELISA kit?
We thank the Reviewer for his observation. The authors have added the reference (A comparative study of commercially available ELISA tests by Alvarez-García et al., 2013) in the corrected version of the manuscript. The authors used IDVet Elisa competitive test that was validated for the detection anti-N. caninum antibodies in serum or plasma from ruminants, dogs, or other susceptible species as reported in the manufacturer’s instructions.
Line 96: “following the manufacturer’s instructions”. Please, you should write a brief description of the protocol (OD, nm, formula for the S/P% and the cut-off).
We thank the Reviewer for his suggestion. The authors have enriched the text with the following information: “The corresponding values for optical density were recorded by a 96-well microplates reader (Spectro Star Omega-BMG Labtech) at a wave length of 450 nm by software Omega 5.70). The results were calculated as the percentage S/N (S/N%). A test result of ≤ 50 S/N% indi-cated a positive result, 50% < S/N% ≤ 60 indicated a doubtful result and > 60 S/N% a negative result. Specific details of the ELISA kits, along with the sensitivities and specificities of the assays, are 100% (IC 95%: 98.8–100%) and 100% (IC 95%: 99.41–100%), respectively”.
Lines 120-124: the whole paragraph contains material and methods and have to be removed from the results section.
We thank the Reviewer for his suggestion. The authors have moved the sentences contains material and methods to the appropriate paragraph.
Lines 128-130: this sentence have to be moved to the discussion section.
We thank the Reviewer for his suggestion. The authors have moved the sentences to the appropriate paragraph.
Lines 130-132: this sentence should be rewrite, placing the seropositive rate at the beginning ot the sentence (149/706). Since the Se and Sp are not 100%, you should calculate the true seroprevalence rate.
We thank the Reviewer for his suggestion. The authors rewrote the sentence.
Table 1: should be deleted. Figure 3 is enough to show the seroprevalence in no-SOF farms. The SOF farms are explained below (consider the option of put SOF and no SOF results in the same section). You should answer this question, is the seroprevalence in SOF farms statistically higher than in no-SOF ones?
We thank the Reviewer for his suggestion. The seroprevalence in SOF farms statistically higher than in no-SOF ones was evaluated by authors. See table 3 in revised manuscript.
“Dubious” better “doubtful”
We thank the Reviewer for his suggestion. The authors have replaced “dubious” with “doubtful”.
Line 153: figure 4 must be removed since data of this pie chart are material and methods (no results).
We thank the Reviewer for his suggestion. The authors have removed figure 4.
Line 150: “the text continues here” should be removed in every part of the manuscript in which the authors wrote it.
We thank the Reviewer for his suggestion. The authors have removed “the text continues here” in revised manuscript.
Lines 157-160: the statistical analysis have to be explained in the material and methods section (and delete this “p-value=0.03018017 so P<0,05” from the results). You should answer this question, are there statistically higher seroprevalence rate in less than two years old cattle from SOF farms compared to non-SOF farms? Lines 161-163: you should also calculate the odds ratio.
We thank the Reviewer for his suggestion. The authors have deleted “p-value=0.03018017 so P<0,05” from the results. Seroprevalence rate in cattle less than two years old from SOF farms compared to non-SOF farms was evaluated by the authors. See table 2 in revised manuscript.
Lines 172-173: “In this study, were tested 10 calf under one year old, which were male or female 172 offspring of N. caninum. positive cows”. This sentence is material and methods (no results).
We thank the Reviewer for his suggestion. The authors have moved the sentences to the appropriate paragraph.
Line 173: maternal antibodies obtained thorough colostrum intake can be detected for several months in the offspring. Please clarify the age of the ofsspring tested and evaluate the possibility of the maternal interference.
We thank the Reviewer for his observation. Subjects were selected between 6-12 months of age to limit the interference of maternal antibodies obtained through colostrum intake as reported in the literature. See Ref n.58.
Line 175: you should detail the variation in the S/P% of these animals along years. It is not enough the maintenance of the seropositivity. Increased S/P% could mean recrudescence of the chronic infection.
We thank the Reviewer for his suggestion. Unfortunately, the authors did not have the necessary number of samples available to respond to this request.
Lines 178-187: all these sentences are not results and they must be moved (avoiding repetition) to the material and methods section.
We thank the Reviewer for his suggestion. The authors have moved all sentences to the appropriate paragraph.
Lines 188-190: X% (13/38) X% (25/38). Youth? Please the English should be corrected by a native speaker.
We thank the Reviewer for his suggestion. The authors have provided corrections and added percentage values.
“This result, evidently higher than that found in the remaining 38 farms examined, is related to the different modes of manifestation of neosporosis” it is not true at all. You must complement your study with additional evaluations to know the predominant route of infection. With your results you only can hypothesize it in the discussion section? And please see if you can do this (are there statistically higher seroprevalence rate in less than two years old cattle from SOF farms compared to non-SOF farms?).
Seroprevalence rate in cattle less than two years old from SOF farms compared to non-SOF farms was evaluated by the authors. See table 2 in revised manuscript.
Lines 192-193: “The presence of dogs was not evidenced on all farms. Among examined farms, were 192 collected blood samples of 21 dogs”. This should be moved to the material and methods section instead of results section.
We thank the Reviewer for his suggestion. The authors have moved all sentences to the appropriate paragraph.
Lines 193-196: if the methology used to test antibodies in dogs has not been validated these results are invalid. In my opinion you should remove the serology of the dogs from the manuscript. The dog seropositivity is not indicative of horizontal transmission since many sero+ dogs have never shed oocysts.
We thank the Reviewer for his observation. The mythology used to test antibodies in dogs is validated by IDVet. The authors believe that it is appropriate to study the seropositivity of dogs because the infection of cattle starts with the dog, although it proceeds and spreads independently of the dog via the vertical transmission route.
Lines 207-217: rewrite to improve the understanding. Many ideas without a guiding line (avoid repetition of information). You must compare you results on the seroprevalence of Neospora in South Italy with previous studies in this country, taking into account the different methologies used.
Lines 210-211: correct citation of the references.
We thank the Reviewer for his suggestion. The authors have revised and updated the bibliography.
Lines 205-217: your hypothesis of horizontal tranmssion should be confirm with solid data/test/analysis.
We thank the Reviewer for his suggestion. In the revised manuscript, the authors focused on vertical transmission, not forgetting the risk of horizontal transmission due to ingestion by livestock of oocysts excreted in faeces by definitive hosts (dogs, dingoes, coyotes, wolves).
Lines 218-219: influence of maternal antibodies in the offspring results?
Subjects were selected between 6-12 months of age to limit the interference of maternal antibodies obtained through colostrum intake as reported in the literature. See Ref n.58.
Lines 241-262: please see my recommendations about dog serology.
We thank the Reviewer for his observation.
Lines 273-278: you did not apply any control measure so this must be removed from the manucript.
We thank the Reviewer for his observation. The authors have removed the sentences.
Lines 279-289: many mixed and inconsistent ideas. Please rewrite.
We thank the Reviewer for his observation.
Line 293: farm seropositivity not seropositivity level.
We thank the Reviewer for his observation.
Lines 296-299: you did not show the number of aborted/repeteadly aborted animals, so this should be removed.
Line 301: you did not study the abortions and the economic impact and therefore it should be removed.
Lines 303-312: remove since you did not apply any control measure. Please give only conclusions about the serology not about abortions, economic impact and control programs.
We thank the Reviewer for his observation. The authors have removed the sentences. the authors rewrote the conclusion paragraph.
Welcoming Reviewer’s suggestion, we have checked the style of the manuscript and we hope that the quality of English language in the revised manuscript can be considered good.

Round 2
Reviewer 1 Report
I am basically satisfied with the author's modification.
Author Response
We thank the Reviewer for appreciating the work.
Reviewer 2 Report
Line 3 (title): double space between serum and in. Title: I suggest to remove “in canine and bovine serum” Line 8: Change “dogs” by “canids” Line 9: farm in plural “farms”. Line 11: Why you use Southern here and South in the title, please homogenize. Line 15: You should cite explicitly the name of the commercial kit used. Line 16: 21% (X/X). Add the number of tested and positive. Line 17: higher number or higher percentage. Is better higher percentage? Should you say compared to (higher than.. compared to…) Line 17: change “group of cattle” by “animals” Line 18: 42,9 is not correct, better 42.9 with a point. Line 21: Keywords: better Italian “cattle” farms Line 24: remove (N. caninum) Line 25: the host range for N. caninum is not wide. Toxoplasma gondii has a wide host range but not N. caninum. Line 25: between parenthesis only cite the intermediate hosts not dogs Line 28: term of gestation. Please edit the English again “to the end of gestation” Line 29: Neospora in Italic letters Lines 29-31: Delete the sentence “other infections… common abortifacient” since it not a neccesary information for your manuscript. Lines 31-33: Delete “ N. caninum has been… parasite in cattle” and say worldwide in the line 24. Lines 35-36: delete “mainly because.. is not yet available” since it is not a needed information for your article and you did not investigate new drugs or vaccines. Line 39: please use good terminology: not modes of reproduction. Please change it by routes of transmission. Line 43: pigs and birds are not important intermediate hosts at all. Please delete it. Lines 45-46: delete “N. caninum is generally considered… endemic or epidemic abortions” because it is duplicate information. Lines 46-49: delete “Most infected cows.. determines the outcome of pregnancy” since most of it is duplicate information (see lines 28-29) and the clinical signs in cattle are not important in the introduction of your manuscript. Lines 50-57: Delete “As reported.. level of infection the a herd” (to many sentences). You are not studying the clinical consequences for the pregnancy and it is not a book so please focus your introduction in available serological assays and previous studies in your country. Line 59: pathognomonic? It is not true because T. gondii cause similar lesions. Lines 62-63. Say something (advantages, disadvantages) about other serological tecniques (WB, IFAT, MAT, ETC) Line 64: Figure 1, delete the text from the top of the figure “Horizontal (A)…” and put it in the legend. Lines 66-71: delete from the legend “Horizontal (postnatal….. via the placenta” since it has been explained in the text. Please not duplicate information. Line 74: northern or Northern? Line 76: Neospora in italics Line 76: South or Southern? (the same for the title of the manuscript) Line 79: Why do you consider it a “representative” number of farms? Why not a simple size calculation? Line 89: add N. caninum “N. caninum infection” Line 91: please not use the word outbreak which for N. caninum is only used in cases of oocyst infection. Better abortion? Lines 102-103: please use the abreviations SOF and no-SOF. Line 112: “did´nt” please edit the English in the whole manuscript (did not) Line 116: please use better xg than rpm to say the centrifugation speed. Line 119: Europa. European? Line 119-124: please include the number of the specific authorization of your protocol and summarize these lines. Line 127: Which kits? Are more than one different kits? Line 128: add competition in the name of the ELISA since you say in the autor replies that the ELISA was the commercial competitive ELISA Line 143: add the statistically significant P value here. Line 169: remove the figure and add “results” Line 176: remove “between 2017 and 2021” since it is material and methods Line 191: not use the Word “outbreak” because you dont know if it was caused by oocysts (try to use abortion instead of outbreak) Line 177-179: I dont undestand this sentence at all “In this study..suspected outbreak”. In any case it is material and methods. You must clarify this sentence. Lines 181-182: cite the figure 3ª and delete the following “ the data presented…. were identified” Trying to avoid duplicate information. Line 186: “6-12 months old male or female” is material and methods and you must delete it from the results. Line 197: Remove dairy cattle samples from the top of the figure and include it in the legend (not in duplicate). Line 201-202: It is material and methods. Please, remove it from these lines since it is the legend of a figure with results. Line 206-209: delete and move it yo material and methods. Also you shouls edit the English (was´nt. Was not?) Line 211: figure. Figure? Lines 211-214: figure 4: …. Please improve the writing. Line 213: different font style Line 215: remove dog samples from the top of the figure. Line 261: remove the figure. Lines 273-274: explain why the seropositivity increases with the age.. possible horizontal transmission? Dog seropositivity? Try to correlate dog and cattle seropositivity in the discussion, avoiding unconnected ideas. Lines 279-287. “However….succesive pregnancies (several sentences). These sentences must be removed since they are not related to your manuscript. Line 301. Modes of reproduction. Better routes of transmission Lines 298-312: improve the writing for a better understanding Line 332: souther or Southern?
Author Response
We thank the auditor for all his comments and suggestions.
- Line 3 (title): double space between serum and in. Title: I suggest to remove “in canine and bovine serum”.
If the article will be accepted for publication, the title will be changed: “Detection of anti-Neospora caninum antibodies on dairy cattle farms in South Italy”.
- Line 8: Change “dogs” by “canids”.
The authors have changed it.
- Line 9: farm in plural “farms”
The authors have changed it.
- Line 11: Why you use Southern here and South in the title, please homogenize.
The authors have homogenized using South Italy.
- Line 15: You should cite explicitly the name of the commercial kit used.
The sentence has been improved.
- Line 16: 21% (X/X). Add the number of tested and positive.
The authors have added the number of tested and positive for cows and dogs.
- Line 17: higher number or higher percentage. Is better higher percentage? Should you say compared to (higher than.. compared to…).
The sentence has been improved.
- Line 17: change “group of cattle” by “animals”
The authors have changed it.
- Line 18: 42,9 is not correct, better 42.9 with a point.
The authors have corrected it.
- Line 21: Keywords: better Italian “cattle” farms
The authors have improved it.
- Line 24: remove (N. caninum)
The authors have removed it.
- Line 25: the host range for N. caninum is not wide. Toxoplasma gondii has a wide host range but not N. caninum.
The sentence has been improved.
- Line 25: between parenthesis only cite the intermediate hosts not dogs
The sentence has been improved.
- Line 28: term of gestation. Please edit the English again “to the end of gestation”
The authors have corrected it.
- Line 29: Neospora in Italic letters
The authors have corrected it.
- Lines 29-31: Delete the sentence “other infections… common abortifacient” since it not a neccesary information for your manuscript.
The authors have deleted the sentence and updated the bibliography.
- Lines 31-33: Delete “ N. caninum has been… parasite in cattle” and say worldwide in the line 24.
The sentence has been improved.
- Lines 35-36: delete “mainly because.. is not yet available” since it is not a needed information for your article and you did not investigate new drugs or vaccines.
To the best of their knowledge, the authors believe that information is appropriate because, at present, prevention is the only way forward.
- Line 39: please use good terminology: not modes of reproduction. Please change it by routes of transmission.
The authors have corrected it.
- Line 43: pigs and birds are not important intermediate hosts at all. Please delete it.
The sentence has been improved.
- Lines 45-46: delete “N. caninum is generally considered… endemic or epidemic abortions” because it is duplicate information. Lines 46-49: delete “Most infected cows.. determines the outcome of pregnancy” since most of it is duplicate information (see lines 28-29) and the clinical signs in cattle are not important in the introduction of your manuscript. Lines 50-57: Delete “As reported.. level of infection the a herd” (to many sentences). You are not studying the clinical consequences for the pregnancy and it is not a book so please focus your introduction in available serological assays and previous studies in your country. Line 59: pathognomonic? It is not true because T. gondii cause similar lesions.
The authors have revised and rewritten the paragraph. The related bibliography has been updated and implemented.
- Lines 62-63. Say something (advantages, disadvantages) about other serological tecniques (WB, IFAT, MAT, ETC)
The authors the authors implemented the techniques section and updated the bibliography.
- Line 64: Figure 1, delete the text from the top of the figure “Horizontal (A)…” and put it in the legend.
The authors have corrected it.
- Lines 66-71: delete from the legend “Horizontal (postnatal….. via the placenta” since it has been explained in the text. Please not duplicate information.
The authors have updated the caption.
- Line 74: northern or Northern?
Northern
- Line 76: Neospora in italics
The authors have corrected it.
- Line 76: South or Southern? (the same for the title of the manuscript)
The authors have corrected Southern in the manuscript.
- Line 79: Why do you consider it a “representative” number of farms? Why not a simple size calculation?
The authors have reformulated the sentence.
- Line 89: add N. caninum “N. caninum infection”
The sentence has been implemented.
- Line 91: please not use the word outbreak which for N. caninum is only used in cases of oocyst infection. Better abortion?
The authors have labelled the dairy farms as: high incidence of reproductive disorders farm (RDF) and no relevant evidence of reproduction problems farm (no-RDF).
- Lines 102-103: please use the abreviations SOF and no-SOF.
The authors have corrected it.
- Line 112: “did´nt” please edit the English in the whole manuscript (did not)
The authors have corrected it.
- Line 116: please use better xg than rpm to say the centrifugation speed.
The authors have implemented the information.
- Line 119: Europa. European?
The authors have corrected it (European)
- Line 119-124: please include the number of the specific authorization of your protocol and summarize these lines.
This paragraph (with all references) as a request from the editor has been added.
- Line 127: Which kits? Are more than one different kits?
The authors have changed it (one type of kit was used).
- Line 128: add competition in the name of the ELISA since you say in the autor replies that the ELISA was the commercial competitive ELISA
The ELISA kit definition was implemented.
- Line 143: add the statistically significant P value
The sentence was implemented.
- Line 169: remove the figure and add “results”
Which figure does the reviewer refer to?
- Line 176: remove “between 2017 and 2021” since it is material and methods
The authors have corrected it.
- Line 191: not use the Word “outbreak” because you dont know if it was caused by oocysts (try to use abortion instead of outbreak)
See reply to line 91
- Line 177-179: I dont undestand this sentence at all “In this study..suspected outbreak”. In any case it is material and methods. You must clarify this sentence.
The sentence has been clarified in the revised manuscript
- Lines 181-182: cite the figure 3ª and delete the following “ the data presented…. were identified” Trying to avoid duplicate information.
The sentence has been improved.
Line 186: “6-12 months old male or female” is material and methods and you must delete it from the results.
The sentence has been moved to “materials and methods” (line 113) of the revised manuscript.
- Line 197: Remove dairy cattle samples from the top of the figure and include it in the legend (not in duplicate).
The figure has been modified
- Line 201-202: It is material and methods. Please, remove it from these lines since it is the legend of a figure with results.
The caption has been modified
Line 206-209: delete and move it yo material and methods. Also you shouls edit the English (was´nt. Was not?)
The sentence has been corrected and moved to “materials and methods” (line115-119) of the revised manuscript.
- Line 211: figure. Figure?
Figure
- Lines 211-214: figure 4: …. Please improve the writing.
The authors have improved the sentence.
- Line 213: different font style
The fonts of the sentence have been standardised.
- Line 215: remove dog samples from the top of the figure.
The figure has been modified.
- Line 261: remove the figure.
Which figure does the reviewer refer to?
- Lines 273-274: explain why the seropositivity increases with the age.. possible horizontal transmission? Dog seropositivity? Try to correlate dog and cattle seropositivity in the discussion, avoiding unconnected ideas.
The authors confirmed that the probability of an animal being positive for N. caninum increases with age due to the increased risk of infection linked to horizontal transmission as also reported in the literature (see ref.16,39, 62-64).
- Lines 279-287. “However….succesive pregnancies (several sentences). These sentences must be removed since they are not related to your manuscript.
The authors have reformulated the sentences.
- Line 301. Modes of reproduction. Better routes of transmission
The authors changed "modes of reproduction” to " routes of transmission” in the revised manuscript.
- Lines 298-312: improve the writing for a better understanding
The authors improved the exposition for a better understanding of the sentence.
- Line 332: souther or Southern?
The authors have written Southern the revised manuscript.

Round 3
Reviewer 2 Report
The authors had good results but they should improve several parts of the manuscript. Below you can find my comments.
Line 16: add competition “IDvet Nc competition”. https://www.id-vet.com/produit/id-screen-neospora-caninum-competition/
Line 25: “can have” better than “have”
Line 30: Neospora in italic letters. Revise it in the whole manuscript.
Line 43: routes in plural.
Lines 43-55. Please, clarify this paragraph. You first say the information about routes of transmission, then about reproduction, then again information about transmission. Tip: not start the paragraph like you did it.
Lines 68-69: delete ISCOM ELISA and only say ELISA because ISCOM is a type of ELISA.
Line 69: I disagree that IFAT is the most reliable tecnique. You can say “one of the most”. Add time comsumption as a cons for IFAT.
Linee 56-57: delete molecular genetics since it has no relation with your study.
Line 63-66: delete the information about IHQ since it is not relevant for your study.
Line 80-83: The authors must improve the writing of this sentence. Also say seropositivity instead of serological positivity. Tip: delete the beginning of the sentence and say directly the % of detection of the parasite in aborted fetuses.
Line 85: use the abrevation (IFAT)
Line 87: inter-farm prevalence. Which is the meaning of inter-farm prevalence, please clarify the terminology in the manuscript.
Line 91. Point at the end of the paragraph.
Lines 98-100. You say that you sampled animals under 2 years and the reasons, but you did not say that you also sampled animals over 2 years and the reason (statistics).
Material and methods: put the institutional animal care permission at the beginning of the material and methods, before the sampling procedures.
Line 112: could you differentially mark in the figure 2 the farms with reproductive disorders?
Lines 122 and 124: change positive by seropositive
Line 125: “these subjects…” for a better understanding, include it in the previous sentence.
Lines 129-130: delete the information between brackets about dogs since it is unneccesary.
Lines 132-135: vacuum tubes? Brand?
Line 146: add competition “IDvet Neospora caninum competition”
Line 156: delete one missing bracket.
Line 162: which statistically significant value did you consider? 0.05?
Line 195-196: the sentence “out off of the 706 samples ..” should be deleted. It is unneccesary.
Line 197-198: “with 34.2%...” the information in this sentence did not correlate with that in the table. Please check it.
Line 200: change “of positive subject” by “born from seropositive subjects”.
Line 201-202: add the S/N% for all the animals along the years (add figure with this information?).
Line 204: Table 1, add the percentages between brackets of the animals in each category. 262 (37.1%).
Line 213: the figure 3 is complete unneccesary because all this information is also in the table I. The authors must delete the figure 3.
Lines 219-222: this information about the dogs is material and methods, no results. In addition, this information should be checked to confirm that it is correct.
Line 224: the results about dogs should be placed in another section (3.2) and change the name of the section 3.1 (only cattle instead of cattle farms).
Line 234: I totally disagree in putting the results from the statistics in a separate result section. Please, put these results as subsections inside the section 3.1 (cattle serology). Also you should clarify, information between brackets, what is each group for a better understanding.
Lines 246-260: add the percentages of the total between brackets (X%) in all the results in these 3 tables. As example 44 (X%) 216 (X%)
Line 264: better infection than parasitosis.
Lines 278-286: you must clearly reduce the information of these sentences since in your study you did not study the outcome for the pregnancies.
Line 295: modes of manifestation? Please change by routes of transmission.
Lines 295-306: these are only hypothesis and you did not deep in the manuscript to clarify the main route of transmission in each farm. You should be more concise (reduce information), but trying to add references to every sentence. Try to put the limitations of your study in the discussion (no outcome for pregnancy, no avidity test… etc).
Line 306: you did not discuss the persistance of the seropositivity in the animals sampled over years. Please add it.
Line 309: change reproductive cicle by life cycle and better delete the beginning of this sentence (is obvious).
Line 316-318: improve the writing of the sentence. It is known that the dog is not needed for the N. caninum propagation.
Line 324: Your study seems not to be the first in South Italy. Please see lines 89-90 of your manuscript.
Lines 327-330. You only were focused on RDF farms but not say anything in the conclusión about the presence in no RDF and the statistics you carried out. Please add it.
Author Response
Replies to Referee 2
We thank the Reviewer for all his comments and suggestions.
The authors had good results but they should improve several parts of the manuscript. Below you can find my comments.
- Line 16: add competition “IDvet Nc competition”. https://www.id-vet.com/produit/id-screen-neospora-caninum-competition/
The authors have added “competition” in the sentence.
- Line 25: “can have” better than “have”
The authors have changed it.
- Line 30: Neospora in italic letters. Revise it in the whole manuscript.
The authors have changed it and the manuscript has been updated.
- Line 43: routes in plural.
The authors have changed it.
- Lines 43-55. Please, clarify this paragraph. You first say the information about routes of transmission, then about reproduction, then again information about transmission. Tip: not start the paragraph like you did it.
The authors have reformulated the sentence and updated the bibliography
- Lines 68-69: delete ISCOM ELISA and only say ELISA because ISCOM is a type of ELISA.
The authors have deleted it
- Line 69: I disagree that IFAT is the most reliable tecnique. You can say “one of the most”. Add time comsumption as a cons for IFAT.
The authors have corrected it.
- Linee 56-57: delete molecular genetics since it has no relation with your study
The authors have corrected it.
- Line 63-66: delete the information about IHQ since it is not relevant for your study.
The authors have corrected it.
- Line 80-83: The authors must improve the writing of this sentence. Also say seropositivity instead of serological positivity. Tip: delete the beginning of the sentence and say directly the % of detection of the parasite in aborted fetuses.
The sentence has been improved.
- Line 85: use the abrevation (IFAT)
The authors used the abrevation (IFAT)
- Line 87: inter-farm prevalence. Which is the meaning of inter-farm prevalence, please clarify the terminology in the manuscript.
The authors explained the term “inter-farm”
- Line 91. Point at the end of the paragraph.
The authors added it
- Lines 98-100. You say that you sampled animals under 2 years and the reasons, but you did not say that you also sampled animals over 2 years and the reason (statistics)
The authors have reformulated the sentence
- Material and methods: put the institutional animal care permission at the beginning of the material and methods, before the sampling procedures.
The authors have moved “the institutional animal care permission” at the beginning of the material and methods, before the sampling procedures.
- Line 112: could you differentially mark in the figure 2 the farms with reproductive disorders?
The authors implemented with two markers the figure 2.
- Lines 122 and 124: change positive by seropositive
The authors have corrected it.
- Line 125: “these subjects…” for a better understanding, include it in the previous sentence.
The authors have updated the sentence
- Lines 129-130: delete the information between brackets about dogs since it is unneccesary.
The authors have corrected it.
- Lines 132-135: vacuum tubes? Brand? Cambiato termine e testo da mettere in risposta
The authors have used BD Vacutainer-CAT (Clot Activator Tube) 10ml. The authors have updated the sentence
- Line 146: add competition “IDvet Neospora caninum competition”
The authors have added “competition” in the sentence.
- Line 156: delete one missing bracket.
The authors have corrected it.
- Line 162: which statistically significant value did you consider? 0.05?
The authors have updated the sentence. They have considered statistically significant value 0.01 and 0.05 based on the conditions described in the text.
- Line 195-196: the sentence “out off of the 706 samples ..” should be deleted. It is unneccesary.
The authors have corrected it.
- Line 197-198: “with 34.2%...” the information in this sentence did not correlate with that in the table. Please check it.
There are 25 positive cattle under two years of age among the total of 39 animals under two years of age (25 +13+1) as shown in the attached table.
· Dairy cattle sample · in RDF |
· Positive to ELISA |
· Negative to ELISA |
· Doubtful to ELISA |
· < 2 years |
· 25 |
· 13 |
· 1 |
· > 2 years |
· 13 |
· 8 |
· 0 |
In the text, statistics for the two RDFs have been added to give a clearer and more complete overview.
- Line 200: change “of positive subject” by “born from seropositive subjects”.
The authors have corrected it.
- Line 201-202: add the S/N% for all the animals along the years (add figure with this information?).
The authors studied ten cows for three years and these subjects remained positive throughout. So, the figure showed low graphical interest.
- Line 204: Table 1, add the percentages between brackets of the animals in each category. 262 (37.1%).
The authors have updated the table 1.
- Line 213: the figure 3 is complete unneccesary because all this information is also in the table I. The authors must delete the figure 3.
The authors have removed the figure 3.
- Lines 219-222: this information about the dogs is material and methods, no results. In addition, this information should be checked to confirm that it is correct.
The authors checked and moved the information on dogs in materials and methods.
- Line 224: the results about dogs should be placed in another section (3.2) and change the name of the section 3.1 (only cattle instead of cattle farms).
The authors have created a second section (3.2) for the results about dogs.
- Line 234: I totally disagree in putting the results from the statistics in a separate result section. Please, put these results as subsections inside the section 3.1 (cattle serology). Also you should clarify, information between brackets, what is each group for a better understanding.
The authors have putted the results from the statistics as subsections (3.1.1) inside the section 3.1
- Line 264: better infection than parasitosis.
The authors have corrected it.
- Lines 278-286: you must clearly reduce the information of these sentences since in your study you did not study the outcome for the pregnancies.
The authors have reduced the information of these sentences
- Line 295: modes of manifestation? Please change by routes of transmission.
The authors have corrected it.
- Lines 295-306: these are only hypothesis and you did not deep in the manuscript to clarify the main route of transmission in each farm. You should be more concise (reduce information), but trying to add references to every sentence. Try to put the limitations of your study in the discussion (no outcome for pregnancy, no avidity test… etc).
- Line 306: you did not discuss the persistance of the seropositivity in the animals sampled over years. Please add it.
The authors added the required information in the text
- Line 309: change reproductive cicle by life cycle and better delete the beginning of this sentence (is obvious).
The authors have updated the sentence
- Line 316-318: improve the writing of the sentence. It is known that the dog is not needed for the N. caninum propagation.
- Line 324: Your study seems not to be the first in South Italy. Please see lines 89-90 of your manuscript.
The authors clarified the sentence in the conclusions.
- Lines 327-330. You only were focused on RDF farms but not say anything in the conclusión about the presence in no RDF and the statistics you carried out. Pleaseaddit.
The authors added the required information
